# Male rats emit aversive 44-kHz ultrasonic vocalizations during prolonged Pavlovian fear conditioning

**Krzysztof Hubert Olszyński**[1†], **Rafał Polowy**[1†], **Agnieszka Diana Wardak**[1], **Izabela Anna Łaska**[1], **Aneta Wiktoria Grymanowska**[1], **Wojciech Puławski**[2], **Olga Gawryś**[3], **Michał Koliński**[2], **Robert Kuba Filipkowski**[1]*

[1]Behavior and Metabolism Research Laboratory, Mossakowski Medical Research Institute, Polish Academy of Sciences, Warsaw, Poland; [2]Bioinformatics Laboratory, Mossakowski Medical Research Institute, Polish Academy of Sciences, Warsaw, Poland; [3]Department of Renal and Body Fluid Physiology, Mossakowski Medical Research Institute, Polish Academy of Sciences, Warsaw, Poland

**\*For correspondence:** rfilipkowski@imdik.pan.pl

[†]These authors contributed equally to this work

**Competing interest:** The authors declare that no competing interests exist.

**Preprint posted** 08 April 2023

**Sent for Review** 04 May 2023

**Reviewed preprint posted** 23 June 2023

**Reviewed preprint revised** 26 April 2024

**Reviewed preprint revised** 04 October 2024

**Version of Record published** 10 December 2024

## eLife Assessment

This **valuable** study investigated the appearance of ultrasonic vocalizations around 44 kHz that occurs in response to prolonged fear conditioning in male rats. Evidence in support of the conclusions is **solid** and may be of interest to some researchers also investigating distress-related ultrasonic vocalizations.

**Abstract** Rats are believed to communicate their emotional state by emitting two distinct types of ultrasonic vocalizations. The first is long '22-kHz' vocalizations (>300 ms, <32-kHz) with constant frequency, signaling aversive states, and the second is short '50-kHz' calls (<150 ms, >32 kHz), often frequency-modulated, in appetitive situations. Here, we describe aversive vocalizations emitted at a higher pitch by male Wistar and spontaneously hypertensive rats (SHR) in an intensified aversive state – prolonged fear conditioning. These calls, which we named '44-kHz' vocalizations, are long (>150 ms), generally at a constant frequency (usually within 35–50-kHz range) and have an overall spectrographic image similar to 22-kHz calls. Some 44-kHz vocalizations are comprised of both 22-kHz-like and 44-kHz-like elements. Furthermore, two separate clustering methods confirmed that these 44-kHz calls can be separated from other vocalizations. We observed 44-kHz calls to be associated with freezing behavior during fear conditioning training, during which they constituted up to 19.4% of all calls and most of them appeared next to each other forming uniform groups of vocalizations (bouts). We also show that some of rats' responses to the playback of 44-kHz calls were more akin to that of aversive calls, for example, heart rate changes, whereas other responses were at an intermediate level between aversive and appetitive calls. Our results suggest that rats have a wider vocal repertoire than previously believed, and current definitions of major call types may require reevaluation. We hope that future investigations of 44-kHz calls in rat models of human diseases will contribute to expanding our understanding and therapeutic strategies related to human psychiatric conditions.

## Introduction

Charles Darwin wrote: "That the pitch of the voice bears some relation to certain states of feeling is tolerably clear" (*Darwin, 1872*). This has also been tolerably clearly observed and widely described

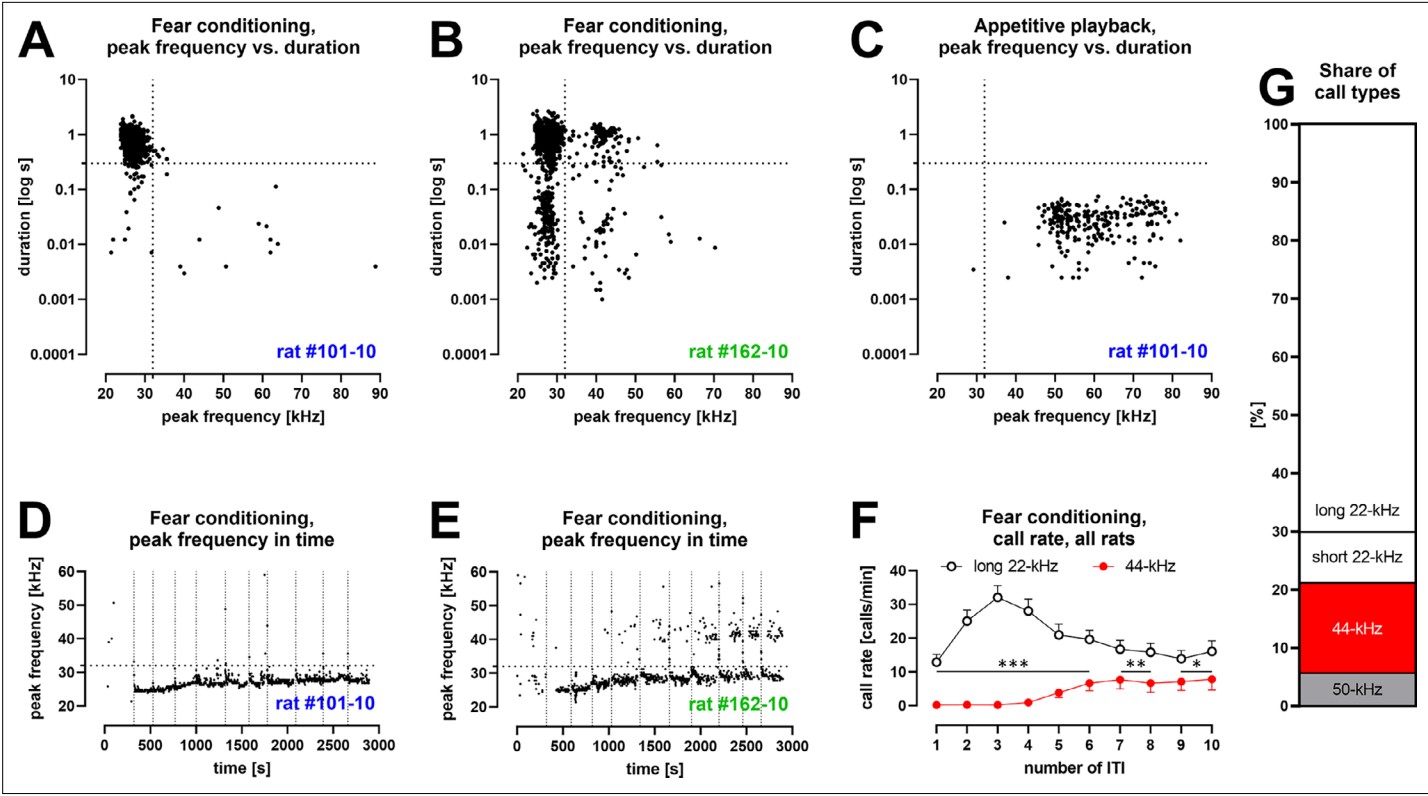

**Figure 1.** Characteristics of vocalizations emitted by Wistar rats during fear conditioning training with 10 aversive foot-shocks (*Table 1*/Exp. 1–3/#2,4,8,13; n = 46). (**A**) Some rats produced aversive 22-kHz vocalizations with typical features, that is, constant-frequency of <32 kHz, >300 ms duration – both values marked as dotted lines); example emission from one rat. (**B**) Some rats produced 44-kHz vocalizations with constant frequency of >32 kHz and long duration (>150 ms); example emission from one rat. (**C**) Rats that emitted aversive vocalizations during fear session produced 50-kHz vocalizations during appetitive playback session the following day (full data published in *Olszyński et al., 2021*); representative data from same rat in (**A**). (**D**) The onset of long 22-kHz alarm calls typically occurred after first shock stimulus (vertical dotted lines mark time of shock deliveries in **D, E**); note the gradual rise in peak frequency, not exceeding 32-kHz (horizontal dotted line in **D, E**); data from the same rat as (**A, C**). (**E**) In rats that emitted 44-kHz calls, the onset was usually delayed to after several foot-shocks; note the gradual rise in peak frequency of both long 22-kHz and 44-kHz vocalizations throughout training (comp. *Figure 1—figure supplement 2C and D*); data from same rat in (**B**). (**F**) Call rate of long 22-kHz calls was higher than 44-kHz calls (*p<0.05, **p<0.01, ***p<0.001) and with different time course – maximum number of 22-kHz calls at inter-trial interval (ITI)-3 (higher than ITI-1, 2, 5–10; <0.0001–0.0005 p levels); and higher number of 44-kHz calls at ITI-5–10, that is, 6.6 ± 2.3 vs. ITI-1–4, that is, 0.4 ± 0.2; p<0.0001; all Wilcoxon; numbers of ITI correspond to the numbers of previous foot-shocks, values are means ± SEM. (**G**) Long 44-kHz vocalizations had a higher incidence rate (15.5%) than short 22-kHz (8.8%) and 50-kHz calls (5.6%); values are calculated for the sum of all vocalizations obtained during the entire training sessions (there were fewer 50-kHz calls, i.e., 3.7%, when vocalizations prior to the first shock were not included). (**A–E**) Dots reflect specified single rat values. (**F, G**) n = 46, other results from these rats are previously published (*Olszyński et al., 2021*; *Olszyński et al., 2023*).

The online version of this article includes the following video and figure supplement(s) for figure 1:

**Figure supplement 1.** Variations of call frequency shown in relation to call duration in Wistar rats that had undergone 6 or 10 trials of delay fear conditioning training (n = 16, selected from *Table 1*/Exp. 2–3/#7,8,13).

**Figure supplement 2.** Changes in distribution (**A, B**), frequency (**C**), duration (**D**), and mean power (**E, F**) of long aversive vocalizations throughout fear conditioning training session.

**Figure supplement 3.** Percentage of animals emitting 44-kHz calls (**A, B**) and percentage of 44-kHz calls in all vocalizations (**C, D**) emitted by Wistar rats and spontaneously hypertensive rats (SHR).

**Figure 1—video 1.** Rat transitioning from emitting long 22-kHz calls to emitting 44-kHz calls.

https://elifesciences.org/articles/88810/figures#fig1video1

for ultrasonic vocalizations of rats (*Brudzynski, 2019*; *Brudzynski, 2021*; *Simola and Granon, 2019*) which emit low-pitched aversive calls and high-pitched appetitive calls. The former are '22-kHz' vocalizations (*Figures 1A and 2A*), with 18–32 kHz frequency range, monotonous and long, usually >300 ms, and are uttered in distress (*Brudzynski, 2013*; *Brudzynski, 2019*; *Brudzynski, 2021*; *Simola and Granon, 2019*). The latter are '50-kHz' vocalizations (*Figure 1C*), are relatively short (10–150

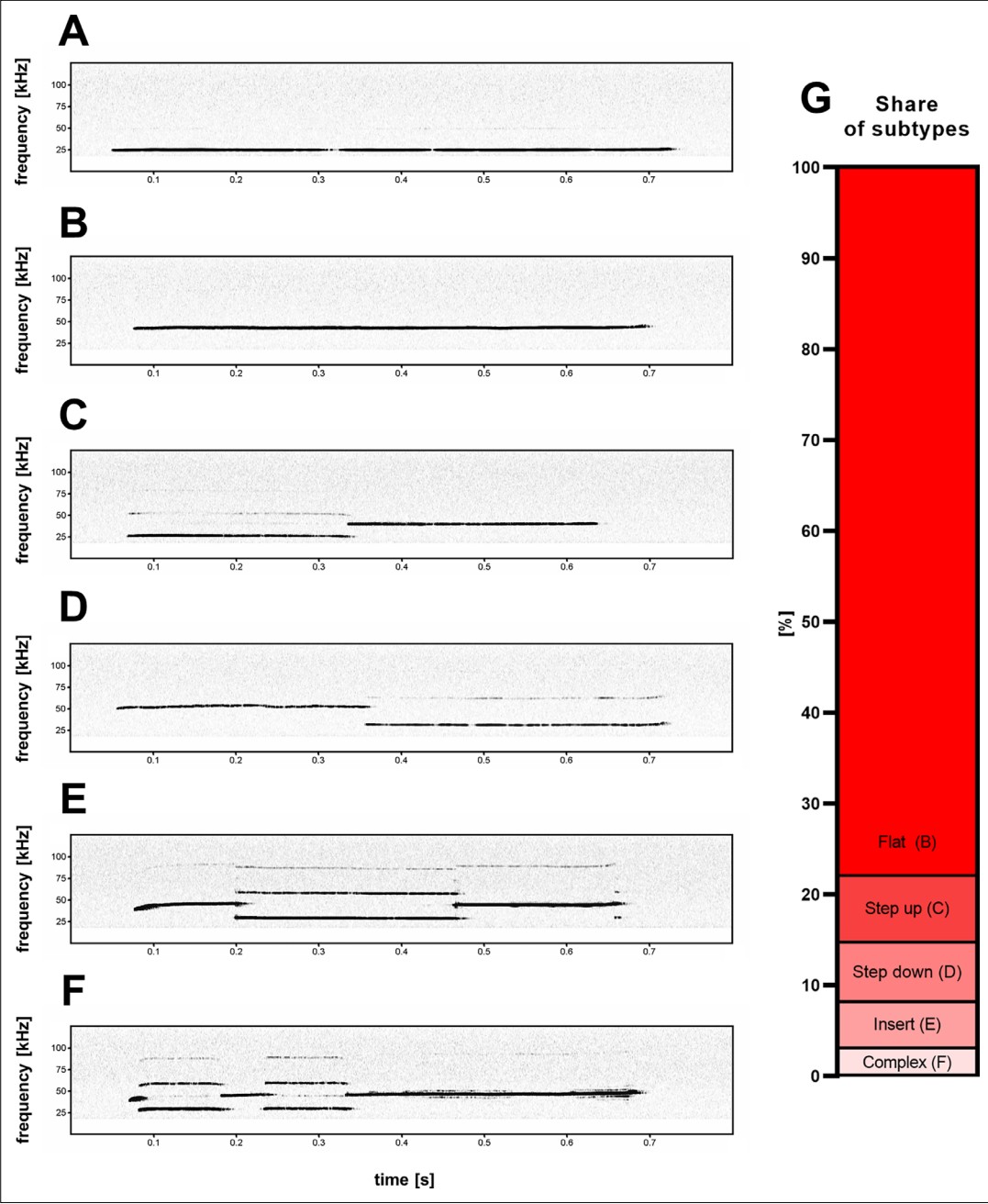

**Figure 2.** Five subtypes (**B–F**) of high-frequency 44-kHz aversive vocalizations. (**A**) Standard aversive 22-kHz vocalization with peak frequency <32 kHz (peak frequency = 24.4 kHz). Five 44-kHz aversive vocalization subtypes: (**B**) flat (constant frequency call; peak frequency = 42.4 kHz); (**C**) step up (peak frequency = 39.5 kHz); (**D**) step down (peak frequency = 52.2 kHz); (**E**) insert (peak frequency = 38.5 kHz); and (**F**) complex (peak frequency = 46.3 kHz). (**G**) Percentage share of 44-kHz call subtypes in all cases of detected 44-kHz vocalizations.

The online version of this article includes the following source data and figure supplement(s) for figure 2:

**Source data 1.** Example of a step up 44-kHz ultrasonic vocalization.

**Source data 2.** Example of a step down 44-kHz ultrasonic vocalization.

**Figure supplement 1.** Non-typical 44-kHz aversive vocalizations.

**Table 1.** All fear conditioning (FC) experiments described in the text.

| # | Exp. | FC type | Strain | # of trials (shocks) | # of rats (n) | Housing | Transmitters | Age in weeks | Description/history |
|---|------|---------|--------|----------------------|---------------|---------|--------------|--------------|---------------------|
| 1 | | | | 0* | 7 | | | | |
| 2 | 1 | Trace | Wistar | 10 | 7 | Single | + | 13 | Rats that had undergone FC after playback experiments (published in *Olszyński et al., 2020*) |
| 3 | | | | 0* | 10 | Paired | | | |
| 4 | | | | 10 | 10 | | | | |
| 5 | | | | 0* | 37 | | | | |
| 6 | | | Wistar | 1 | 16 | | | | |
| 7 | | | | 6 | 22 | | | | |
| 8 | 2 | Delay | | 10 | 19 | Paired | + | 12 | FC experiments which were described before in detail (*Olszyński et al., 2021*; *Olszyński et al., 2023*) |
| 9 | | | | 0* | 31 | | | | |
| 10 | | | SHR | 1 | 17 | | | | |
| 11 | | | | 6 | 17 | | | | |
| 12 | | | | 10 | 15 | | | | |
| 13 | 3 | Delay | Wistar | 10 | 10 | Paired | - | 12 | New FC experiment |

*Control groups.

ms), frequency-modulated, usually within 35–80 kHz, and they signal appetitive and rewarding states (*Simola and Granon, 2019*; *Brudzynski, 2013*; *Brudzynski, 2019*; *Brudzynski, 2021*). Therefore, these two types of calls communicate the animal's emotional state to their social group (*Brudzynski, 2013*). Low-pitch (<32 kHz), short (<300 ms; *Figure 1B*) calls, assumed to also express a negative aversive state, have been described but their role is not clearly established (*Brudzynski, 2013*). Notably, high-pitch (>32 kHz), long and monotonous ultrasonic vocalizations have not yet been described. Here, we show these unmodulated rat vocalizations with peak frequency (i.e., peak frequency of a given vocalization is defined here as the highest power peak in the averaged spectrum of the entire element) at about 44 kHz (*Figure 1B and E*; *Figure 2B*), emitted in aversive experimental situations, especially in prolonged fear conditioning.

## Results
### High, long, and unmodulated calls

In three separate experiments (all summarized in *Table 1*/Exp. 1–3, see 'Materials and methods'), that is, one with trace fear conditioning (*Table 1*/Exp. 1) and two with delay fear conditioning (*Table 1*/Exp. 2–3), one of which has already been described (*Table 1*/Exp. 2, *Olszyński et al., 2021*; *Olszyński et al., 2023*), 53 of all 84 conditioned Wistar rats (*Table 1*/Exp. 1–3/#2,4,6–8,13, *Figure 1B–E*, *Figure 1—figure supplement 1B and C*) displayed vocalizations that were high-pitched, that is, in the range of 50-kHz calls, but long and monotonous (*Figure 2B*). These vocalizations, for example, top-right group in *Figure 1B* and *Figure 1—figure supplement 1C*, were outside the defined range (*Brudzynski, 2019*; *Brudzynski, 2021*; *Simola and Granon, 2019*) for both 50-kHz (bottom-right group in *Figure 1C*, *Figure 1—figure supplement 1A–C*) and 22-kHz calls (top-left group in *Figure 1A and B*, *Figure 1—figure supplement 1A–C*). These vocalizations were also observed in a different rat strain acquired from a different breeding colony, that is, spontaneously hypertensive rats (SHR) (*Brudzynski, 2021*; *Okamoto and Aoki, 1963*), also trained in delay fear conditioning (*Table 1*/Exp. 2/#10–12; *Olszyński et al., 2023*). In particular, 6 of the 49 conditioned SHR displayed high-pitch, long, monotonous vocalizations (e.g., *Figure 2—figure supplement 1G*); moreover, we observed more of these vocalizations in Wistar rats compared to SHR (*Table 1*/Exp. 2/#6–8,10–12) in both training, p<0.0001, and test sessions, p=0.0030, Mann–Whitney.

Overall, we analyzed 140,149 vocalizations from all fear conditioning experiments (*Table 1*/Exp. 1–3/#1–13, n = 218) and through trial-and-error, we set new criteria, namely peak frequency of >32 kHz and >150 ms duration to define the calls described above. We manually verified the results

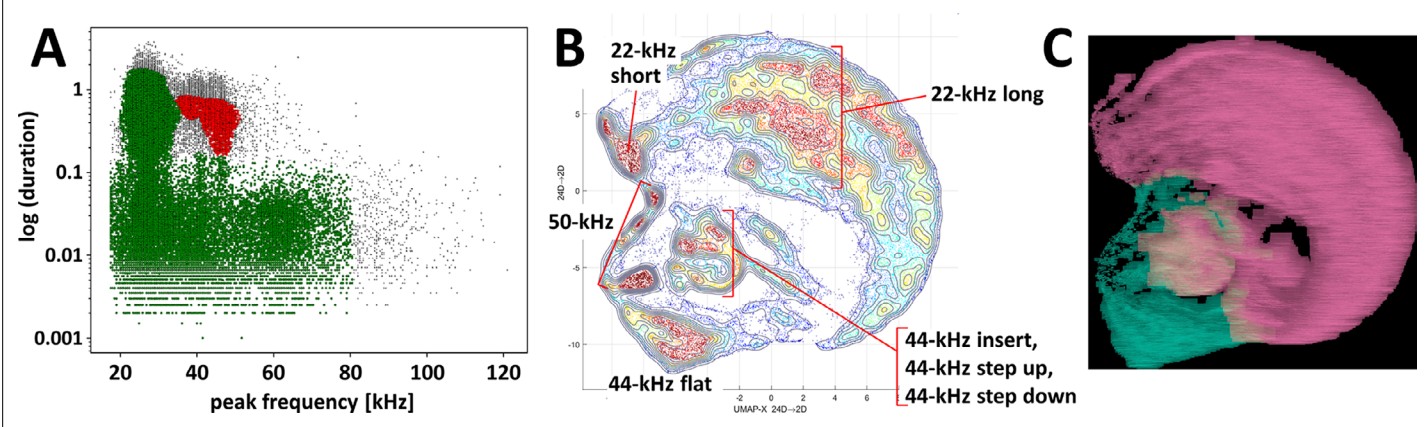

**Figure 3.** Clustering of ultrasonic vocalizations from fear conditioning training sessions using two independent methods. (**A**) DBSCAN algorithm (ε = 0.14) clustering of vocalizations from all fear conditioning experiments (*Table 1*/Exp. 1–3/#1–13, n = 218), silhouette coefficient = 0.198, two clusters emerge, cluster of green dots n = 77,243 (due to high generality of cluster average peak frequency and duration deemed redundant), cluster of red dots n = 5,646 (average peak frequency = 43,826.6 Hz, average duration = 0.524 s), some calls were not assigned to any cluster, that is, outlier vocalizations, black dots, n = 4,139. (**B, C**) Clustering by k-means algorithm and visualization of calls emitted by selected rats, that is, with >30 of 44-kHz vocalizations, during trace and delay fear conditioning training (n = 26, selected from *Table 1*/Exp. 1–3/#2,4,7,8,11–13), total number of calls n = 40,084. (**B**) Topological plot of ultrasonic calls using UMAP embedding, particular agglomerations of calls labeled with their type or subtype. (**C**) Spectrogram images from DeepSqueak software superposed over plot B, colors denote clusters from unsupervised clustering, number of clusters set using elbow optimization (max number = 4), two clusters emerge; see also *Figure 3—figure supplement 1*.

The online version of this article includes the following figure supplement(s) for figure 3:

**Figure supplement 1.** Clustering of ultrasonic vocalizations from rats emitting 44-kHz calls using UMAP and k-means.

**Figure supplement 2.** Examples of parts of ultrasonic bouts comprised of long 22-kHz vocalizations (**A**) and 44-kHz vocalizations (**B**) emitted by Wistar rats during fear conditioning training.

on the spectrogram using these parameters and only 308 calls (0.2%) were incorrectly assigned (i.e., exceptionally long 50-kHz vocalizations misplaced in the newly defined group or borderline-short vocalizations of the newly defined group misplaced to 50-kHz calls). Hence, the new parameters correctly assigned 99.8% of cases and are thus effective to distinguish the newly defined calls in an automated fashion. Finally, 10,445 newly defined calls were identified, which constituted 7.5% of the total calls during fear conditioning experiments (*Table 1*/Exp. 1–3; comp. *Figure 1G*). These vocalizations have a peak frequency range from 32.2 to 51.5 kHz (95% of cases) with an average peak frequency of 42.1 kHz, and they exhibited 43.8 kHz peak frequency at the cluster center in a DBSCAN analysis (*Figure 3A*). In line with the accepted nomenclature convention, underlining the relationship with 22-kHz vocalizations, we christened this newly defined calls as '44-kHz vocalizations'.

## 44-kHz calls in long aversive stimulation

We found 44-kHz vocalizations especially in rats that received multiple electric shocks. When we analyzed all Wistar rats that had undergone 10 trials of fear conditioning (*Table 1*/Exp. 1–3/#2,4,8,13; n = 46), these vocalizations were less frequent following the first trial (1.2 ± 0.4% of all calls), and increased in subsequent trials, particularly after the 5th (8.8 ± 2.8%), through the 9th (19.4 ± 5.5%, the highest value), to the 10th (15.5 ± 4.9%) trials, where 44-kHz calls gradually replaced 22-kHz vocalizations in some rats (*Figure 1F*, *Figure 1—figure supplement 2A and B*, *Figure 1—video 1*; comp. *Figure 1D* vs. *Figure 1E*). Please note that the majority of the 22-kHz calls were emitted after the 3rd shock, that is, during the 3rd inter-trial interval (ITI), while 44-kHz vocalizations were emitted in the second part of the training, that is, 5th to 10th ITI (*Figure 1F*, comp. *Figure 1—figure supplement 2A and B*). From this group of rats (n = 46), n = 41 (89.1%) emitted long 22-kHz calls, and 32 of them (69.6%) emitted 44-kHz calls, that is, every animal that produced 44-kHz calls also emitted long 22-kHz calls (*Figure 1—figure supplement 2A and B*). The prevalence of 44-kHz calls varied greatly among individual rats, such that for n = 3 rats, 44-kHz vocalizations accounted for >95% of all calls during at least one ITI (e.g., 140 of total 142, 222 of 231, and 263 of 265 tallied 44-kHz calls), and in n = 9 rats,

44-kHz vocalizations constituted >50% of calls in more than one ITI. The prevalence of 44-kHz calls in all experimental conditions analyzed in all animal groups is shown in *Figure 1—figure supplement 3*.

Notably, there were more 44-kHz vocalizations during fear conditioning training than testing in all fear-conditioned Wistar rats (*Table 1*/Exp. 1–3/#2,4,6–8,13; n = 84; 3.63 ± 0.99 vs. 0.23 ± 0.13 calls/min; p<0.0001; Wilcoxon).

In a recent publication during this paper's review process, *Gonzalez-Palomares et al., 2023*, in line with the findings reported here, investigated and described 44-kHz vocalizations following prolonged (10-trial protocol) odor fear conditioning. These calls were observed predominantly during the late ITI, that is, 8th–10th ITI (*Gonzalez-Palomares et al., 2023*; Fig. S4C; please note 4th–7th ITI were not investigated) after the shock presentations (Fig. S4B therein), which complement our results.

## Changes in frequency, duration, and mean power of long aversive calls during conditioning

Analyzing Wistar rats that had undergone 10 trials of fear conditioning (*Table 1*/Exp. 1–3/#2,4,8,13; n = 46), we also observed the frequencies of 22-kHz calls to gradually rise throughout fear conditioning training, that is, during subsequent ITI – from 24.5 ± 0.1 to 27.9 ± 0.4 kHz (*Figure 1D and E*, *Figure 1—figure supplement 2C*; p<0.0001, Friedman, p=0.0039, Wilcoxon). The frequency levels of 44-kHz vocalizations also appeared to rise – from 37.8 ± 2.1 to 39.6 ± 1.3 kHz (*Figure 1E*, *Figure 1—figure supplement 2C*) but we were unable to statistically demonstrate it (p=0.0155, Friedman, p=0.0977, Wilcoxon).

There was a shortening of long 22-kHz calls during the first four ITI from 969.6 ± 43.1 ms to 794.6 ± 39.8 ms (p<0.0001, Friedman; p<0.0001, Wilcoxon, *Figure 1—figure supplement 2D*), while 44-kHz vocalizations were longest during the 4th ITI (the time of their substantial appearance, comp. *Figure 1F*), that is, 775.0 ± 135.7 ms, and shortened over subsequent ITI (619.6 ± 58.1 ms for the 10th ITI, *Figure 1—figure supplement 2D*, p=0.0227, Friedman; p=0.0234, Wilcoxon).

**Table 2.** Freezing associated with emission of long, monotonous vocalizations.

All Wistar rats that had undergone 10 trials of fear conditioning training were analyzed (*Table 1*/Exp. 1–3/#2,4,8,13; n = 46). (A) Freezing (%) in 10-s-long bins where rats emitted exclusively long 22-kHz vocalizations vs. exclusively 44-kHz vocalizations. Results were compared to baseline freezing levels before conditioning training ('First 5 min') and during 10-s-long periods with no vocalizations ('no calls'). More information in the text. (B) Freezing during the emission episodes of long 22-kHz and 44-kHz calls. Pairs of 44-kHz and long 22-kHz vocalizations were randomly selected from each animal. Freezing levels (%) did not differ between 22-kHz vs. 44-kHz calls (0.2054–0.7776 p levels, Wilcoxon). Minimum freezing duration used: 30 frames (A), 3 frames (for pairs of ≥150 ms vocalizations), or 5, 10, and 15 frames for ≥500 ms vocalizations (B).

(A) Freezing behavior during 10-s-long time intervals; analyzed with 30 frames

| | | Freezing (%) | | |
| | | 10-s-long bins with | | |
| Group of rats analyzed | First 5 min | no calls | 22-kHz calls only | 44-kHz calls only |
|---|---|---|---|---|
| Rats with long 22-kHz calls (n = 41) | 6.3 ± 2.3 | 39.7 ± 3.5 | 47.8 ± 3.3***,# | NA |
| Rats with long 22-kHz calls and 44-kHz calls (n = 21) | 9.1 ± 4.2 | 33.7 ± 4.9 | 40.4 ± 4.3*** | 50.3±7.0***,# |

(B) Freezing behavior during selected calls; analyzed with reduced number of frames

| Rats with long 22-kHz calls and 44-kHz calls | | Freezing (%) during emission of | |
| | No. of frames | 22-kHz calls | 44-kHz calls |
|---|---|---|---|
| with calls of ≥150 ms duration (n = 32) | 3 | 49.5 ± 7.6 | 58.8 ± 8.0 |
| | 5 | 67.3 ± 7.3 | 54.1 ± 9.1 |
| | 10 | 61.9 ± 8.1 | 53.3 ± 9.1 |
| with calls of ≥500 ms duration (n = 28) | 15 | 52.6 ± 9.0 | 48.4 ± 9.4 |

NA, not analyzed.

*** vs. 'First 5 min', p<0.001; # vs. 'no calls', p<0.05; both Wilcoxon.

Finally, the sound mean power of 44-kHz vocalizations appeared to remain stable throughout the 10-trial sessions, while during the first half of the training, that is, 1st–5th ITI, 22-kHz calls were not only significantly more frequent but also louder than during the second half, that is, 6th–10th ITI (p<0.0001, Wilcoxon). Consequently, long 22-kHz calls appeared louder than 44-kHz calls (p=0.0397–0.0038, Mann–Whitney). However, in the second half of the session, this difference dissipated due to the diminishing amplitude of 22-kHz vocalizations (p=0.0083, Friedman; p=0.0046, Wilcoxon), while the amplitude of 44-kHz calls remained stable (p=0.0663, Friedman; p=0.2661, Wilcoxon; 6th ITI through 10th ITI for both; *Figure 1—figure supplement 2E*). After adjusting for angle-dependent hardware attenuation (see 'Materials and methods', 'Sound mean power'), the situation reversed (*Figure 1—figure supplement 2F*). Both long 22-kHz and 44-kHz vocalizations showed similar amplitude levels during the first half of the fear conditioning training session, while during the 6th–10th ITI, 44-kHz calls were significantly louder than long 22-kHz calls (p=0.0007–0.0097, Mann–Whitney).

## 44-kHz calls linked to freezing

We investigated the freezing behavior of all Wistar rats emitting 44-kHz vocalizations during 10 trials of fear conditioning training (*Table 1*/Exp. 1–3/#2,4,8,13; n = 46). The training sessions were divided into 10-s-long time bins, from which we analyzed only the bins that had exclusively long 22-kHz or 44-kHz calls. For comparison, we also measured the freezing levels during the first 5 min of the trial (baseline freezing levels before any foot-shocks) as well as the bins in which animals did not vocalize (from the period after the first shock to the end of the session). Of the n = 46 rats analyzed, n = 41 emitted 22-kHz vocalizations, from which n = 32 also emitted 44-kHz vocalizations, from which only n = 21 were determined to have both – 10-s-long bins of 22-kHz calls only and 44-kHz calls only (*Table 2A*). Freezing during the bins of 22-kHz calls only (p<0.0001, for both groups) and during 44-kHz calls only bins (p=0.0003) was higher than during the first 5 min baseline freezing levels of the session. Also, the freezing associated with emissions of 44-kHz calls only was higher than during bins with no ultrasonic vocalizations (p=0.0353), and it was also 9.9 percentage points higher than during time bins with only long 22-kHz vocalizations, but the difference was not significant (p=0.1907; all Wilcoxon).

To further investigate this potential difference, we measured freezing during the emission of randomly selected single 44-kHz and 22-kHz vocalizations. The minimal freezing behavior detection window was reduced to compensate for the higher resolution of the measurements (3, 5, 10, or 15 video frames were used). There was no difference in freezing during the emission of 44-kHz vs. 22-kHz vocalizations for ≥150-ms-long calls (3 frames, p=0.2054) and for ≥500-ms-long calls (5 frames, p=0.2404; 10 frames, p=0.4498; 15 frames, p=0.7776; all Wilcoxon, *Table 2B*).

## 44-kHz calls sorted into five subtypes

While the majority of 44-kHz vocalizations were of continuous unmodulated frequency (*Figure 2B*), some comprised additional elements. Based on the composition of individual call elements and their relation to each other, we manually sorted the calls into five categories (*Figure 2B–F*). If the start (*prefix*) or end (*suffix*) portion of a call was less than 1/5th the length of the following or previous element, this portion of the call was not considered in its categorization into the five subtypes. The names and descriptions of the five subtypes are: **flat** – single element with near constant frequency and little to no interruptions to the sound continuity on the spectrogram; **step up** – two elements with an instantaneous frequency jump, where the first element is of lower frequency; **step down** – two elements with an instantaneous frequency jump, where the first element is of higher frequency; **insert** – three elements with an instantaneous frequency change, where the middle element is of different frequency; and **complex** – more than three elements with instantaneous frequency changes.

## 44-kHz and 22-kHz calls closely related

44-kHz were emitted in aversive behavioral situations – as 22-kHz calls are observed (*Antoniadis and McDonald, 1999*; *Dupin et al., 2019*; *Taylor et al., 2017*). Both types of calls are long (usually >300 ms) and frequency-unmodulated. Some of the elements constituting such as step up; step down; insert and complex 44-kHz vocalizations (*Figure 2C–F*) were at a lower frequency – typical for 22-kHz vocalizations. Vice versa we also observed 22-kHz calls with 44-kHz-like elements. Therefore, we propose that these long 22-kHz and 44-kHz vocalizations constitute a *supertype* group of long unmodulated aversive calls ('long 22/44-kHz vocalizations').

We observed a stable, approximately 1.5 ratio in peak frequency levels between 22-kHz and 44-kHz vocalizations within individual rats. Specifically, in 14 rats (13 Wistar and 1 SHR) with a clear transition from 22-kHz to 44-kHz calls during the fear conditioning training session (n = 14, selected from *Table 1*/Exp. 1–3/#2,4,6–8,10–13), the proportion between the frequencies of the long 22-kHz vocalizations and the long 44-kHz calls was 1.48 ± 0.02. Similar results were obtained for 70 step up (1.53 ± 0.03) and 65 step down (1.59 ± 0.02) 44-kHz calls – altogether suggesting a 1.5 times or 3:2 frequency ratio. This ratio and its relevance have been observed in invertebrates and vertebrates including human speech and music (*Hoeschele, 2017*). In music theory, 3:2 frequency ratio is referred to as a perfect fifth and is often featured, for example, the first two notes of the *Star Wars* 1977 movie (ascending, i.e., step up; comp. *Figure 2C*, *Figure 2—source data 1*) and *Game of Thrones* 2011 television series (descending, i.e., step down; comp. *Figure 2D*, *Figure 2—source data 2*) theme songs. All of these may point to a common basis for this sound interval and its prevalence that could be explained by the observation that all physical objects capable of producing tonal sounds generate harmonic vibrations, the most prominent being the octave, perfect fifth, and major third (*Christensen, 1993*, discussed in *Bowling and Purves, 2015*).

## 44-kHz calls separated in cluster analyses

Next, we showed that 44-kHz calls indeed constitute a separable type of ultrasonic vocalizations as it was sorted into isolated clusters by two different methods. First, using the DBSCAN algorithm method based on calls' peak frequency and duration, we were able to divide all vocalizations recorded during all training sessions into 44-kHz vocalizations vs. all other vocalizations as two separate clusters (*Figure 3A*). Secondly, a clustering algorithm that includes call contours, that is, k-means with UMAP done via DeepSqueak (*Figure 3B and C*, *Figure 3—figure supplement 1*), sorted 44-kHz vocalizations of different subtypes including unusual ones (*Figure 2—figure supplement 1A–F*), into topologically-separate groups. Notably, flat 44-kHz calls were consistently in a separate cluster from 22-kHz calls (*Figure 3C*, *Figure 3—figure supplement 1B*).

## 44-kHz calls formed separate groups

We have examined transition probabilities for call-type transitions within all vocalizations and within vocalization bouts in all Wistar rats during 10 trials of fear conditioning training (*Table 1*/Exp. 1–3/#2,4,8,13; n = 46). Transition probabilities between vocalization types were defined by counting the number of specific pair sequences and dividing by all observed sequence pairs where the first type of signal is followed by any signal. The most probable call following long 22-kHz vocalization was another long 22-kHz vocalization (94.4% probability). Similarly, the most probable call following 44-kHz vocalization was another 44-kHz call (83.1%). These values augmented when the analysis was limited to vocalizations emitted only in bouts, that is, with <320 ms time gaps between calls (*Wöhr et al., 2005*), and reached 96.3 and 86.5%, respectively. Examples of parts of groups of both types of calls are demonstrated in *Figure 3—figure supplement 2*.

## Response to 44-kHz playback

To describe the behavioral and physiological impact of 44-kHz vocalizations, we performed playback experiments in two separate groups of rats ('Materials and methods', *Figure 4*, *Figure 4—figure supplement 1*). Overall, the responses to 44-kHz aversive calls presented from the speaker were either similar to 22-kHz vocalizations or in-between responses to 22-kHz and 50-kHz playbacks. For example, the heart rate of rats exposed to 22-kHz and 44-kHz vocalizations decreased, and increased to 50-kHz calls (*Figure 4A*, comp. *Olszyński et al., 2020*). Whereas the number of vocalizations emitted by rats was the highest during and after the playback of 50-kHz, intermediate to 44-kHz and lowest to 22-kHz playbacks (*Figure 4B and C*, *Figure 4—figure supplement 1E and F*). Additionally, the duration of 50-kHz vocalizations emitted in response to 44-kHz playback was also intermediate, that is, longer than following 22-kHz playback (*Figure 4D*) and shorter than following 50-kHz playback (*Figure 4D*, *Figure 4—figure supplement 1G*). Finally, similar tendencies were observed in the distance traveled and time spent in the half of the cage adjacent to the speaker (*Figure 4—figure supplement 1A–D*).

## Discussion

As Charles Darwin noted above (*Darwin, 1872*) and other researchers have confirmed (*Briefer et al., 2012*), the frequency level of animal calls is a vocal parameter that changes in accordance with its

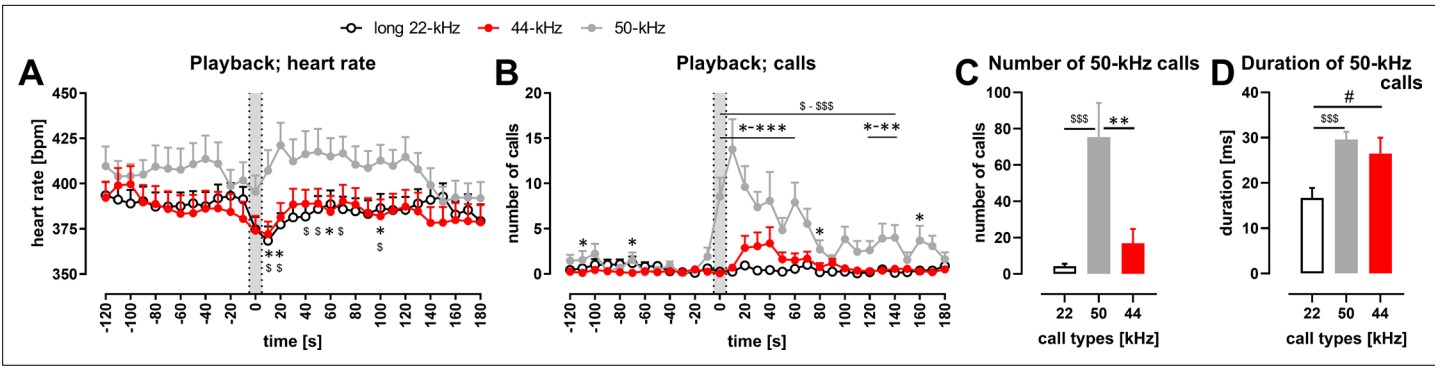

**Figure 4.** Physiological and behavioral response to playback of 44-kHz calls (vs. 50-kHz and 22-kHz calls) presented from a speaker to naïve Wistar rats. (**A**) Heart rate (HR). (**B**) The number of emitted vocalizations. (**A, B**) Gray sections correspond to the 10-s-long ultrasonic playback. Each point is a mean for a 10-s-long time-interval with SEM. (**C, D**) Properties of 50-kHz vocalizations emitted in response to ultrasonic playback, that is, number of calls (**C**) and duration (**D**) calculated from the 0-120 s range. (**A**) 50-kHz playback resulted in HR increase (playback time interval vs. 10–30 s time interval, p=0.0007), while the presentation of the aversive playbacks resulted in HR decrease, both in case of 22-kHz (p<0.0001) and 44-kHz (p=0.0014, average from −30 to −10 time intervals [i.e., 'before'] vs. playback interval, all Wilcoxon), which resulted in different HR values following different playbacks, especially at +10 s (p=0.0097 for 50-kHz vs. 22-kHz playback; p=0.0275 for 50-kHz vs. 44-kHz playback) and +20 s time intervals (p=0.0068, p=0.0097, respectively, all Mann–Whitney). (**B**) 50-kHz playback resulted also in a rise of evoked vocalizations (*before* vs. 10–30 s time interval, p=0.0002, Wilcoxon) as was the case with 44-kHz playback (p=0.0176 in respective comparison), while no rise was observed following 22-kHz playback (p=0.1777). However, since the increase in vocalization was robust in case of 50-kHz playback, the number of emitted vocalizations was higher than after 22-kHz playback (e.g., p<0.0001 during 0–30 time intervals) as well as after 44-kHz playback (e.g., p<0.0001 during 0–10 time intervals, both Mann–Whitney). Finally, when the increases in the number of emitted ultrasonic calls in comparison with *before* intervals were analyzed, there was a difference following 44-kHz vs. 22-kHz playbacks during 30 s and 40 s time intervals (p=0.0420 and 0.0430, respectively, Wilcoxon). (**C**) During the 2 min following the onset of the playbacks, rats emitted more ultrasonic calls during and after 50-kHz playback in comparison with 22-kHz (p<0.0001) and 44-kHz (p=0.0011) playbacks. The difference between the effects of 22-kHz and 44-kHz playbacks was not significant (p=0.2725, comp. *Figure 4—figure supplement 1F*; all Mann–Whitney). (**D**) Ultrasonic 50-kHz calls emitted in response to playback differed in their duration, that is, they were longer to 50-kHz (p=0.0004) and 44-kHz (p=0.0273, both Mann–Whitney) playbacks than to 22-kHz playback. * 50-kHz vs. 44-kHz, $ 50-kHz vs. 22-kHz, # 22-kHz vs. 44-kHz; one character (*, $, or #), p<0.05; two, p<0.01; three, p<0.001; Mann–Whitney (**A, B**) or Wilcoxon (**C, D**). Values are means ± SEM, n = 13–16.

The online version of this article includes the following figure supplement(s) for figure 4:

**Figure supplement 1.** Behavioral response to playback of 44-kHz calls (vs. 50-kHz and 22-kHz calls).

arousal state (intensity) or emotional valence (positive/negative state). The frequency shifts toward both higher and lower levels, that is, alterations were observed during both positive (appetitive) and negative (agonistic/aversive) situations; however, as a general rule, frequency usually increases with an increase in arousal (*Briefer et al., 2012*). We would like to propose a hypothesis that our prolonged fear conditioning increased the arousal of the rats with no change in the valence of the aversive stimuli.

It could also be speculated that several other factors, apart from increased arousal, contributed to the emergence of 44-kHz vocalizations in our fear-conditioned rats, for example, heightened fear, stress/anxiety, annoyance/anger, disgust/boredom, grief/sadness, despair/helplessness, and weariness/fatigue. It is not possible, at this stage, to definitively determine which factors played a decisive role. Please note that the potential contribution of these factors is not mutually exclusive.

However, several arguments support the idea that 44-kHz vocalizations communicate an increased negative emotional state. First, in general, ultrasonic vocalizations serve as a means of communicating rats' emotional state (*Brudzynski, 2013*). Second, the changing of *the pitch of the voice bears some relation to certain states of feeling* (*Darwin, 1872*). Third, 44-kHz calls were notably more frequent during prolonged aversive stimulation, that is, the 5th–10th trials of fear conditioning training. Fourth, they were linked to freezing. Fifth, they appeared as partial replacements of, established as aversive, 22-kHz calls – in the presence of the same painful stimulus. Sixth, numerous instances of vocalizations featured both 22-kHz-like and 44-kHz-like call elements.

Also, several observations contradict the potential contribution of fatigue. The sound mean power of 44-kHz vocalizations was comparable to, or possibly even higher than, that of 22-kHz calls, despite the higher energy costs associated with producing higher-pitched calls (*Sonninen and Hurme, 1998*), that is, the rats emitting 44-kHz calls invested additional energy to communicate their emotional state; both in vivo measurements (*Riede, 2013*) and computer modeling (*Håkansson et al., 2022*)

demonstrated that producing calls of higher frequency, such as 50-kHz vs. 22-kHz, requires increased activity of various muscles. Additionally, the mean power of 44-kHz vocalizations remained strong and stable for several trials – in contrast to 22-kHz vocalizations. Finally, when 44-kHz calls started to appear in significant numbers, that is, after the 4th–5th trials of fear conditioning training, they were as long as 22-kHz vocalizations.

Concerning the latter, we observed a significant decrease in the mean power of 22-kHz vocalizations during the fear conditioning training session. Such reduction could potentially be attributed to fatigue (as observed in humans, *Kitch and Oates, 1994*), despair (e.g., as a reaction to the lack of effects from repeated emissions of 22-kHz calls), or both. The reduction in the amplitude of 22-kHz calls during the 10-trial fear conditioning training was also recently observed by others (*Gonzalez-Palomares et al., 2023*).

Importantly, it has been demonstrated multiple times that training rats with several electric foot-shocks (i.e., 5–10 shocks) produces a qualitatively different kind of fear memory compared to training with only 1–2 shocks. Training with more numerous shocks has been shown to result in augmented freezing (e.g., *Fanselow and Bolles, 1979*; *Haubrich et al., 2020*; *Haubrich and Nader, 2023*; *Poulos et al., 2016*; *Wang et al., 2009*), which reflects a more intense fear memory that is resistant to extinction (*Haubrich et al., 2020*; *Haubrich and Nader, 2023*), resistant to reconsolidation blockade (*Haubrich et al., 2020*; *Wang et al., 2009*; *Finnie and Nader, 2020*), associated with downregulation of NR2B NMDA-receptor subunits as well as elevated amyloid-beta concentrations in the lateral and basal amygdala (*Finnie and Nader, 2020*; *Wang et al., 2009*). Additionally, it involves activation of the noradrenaline-locus coeruleus system (*Haubrich et al., 2020*) and collective changes in connectivity across multiple brain regions within the neural network (*Haubrich and Nader, 2023*).

Notably, it has also been shown that higher freezing as a result of fear conditioning training correlates with increased concentrations of stress hormone, corticosterone, in the blood (*Dos Santos Corrêa et al., 2019*). The rats subjected to 6- and 10-trial fear conditioning training, whose results are reported herein (*Table 1*/Exp. 2/#7,8,11,12; n = 73), also demonstrated higher freezing than rats subjected to 1-trial conditioning (*Table 1*/Exp. 2/#6,10; n = 33), which is reported elsewhere (*Olszyński et al., 2021*, Fig. S1C–E; *Olszyński et al., 2023*, Fig. S1D–G). Therefore, we postulate that emission of 44-kHz calls is associated with increased stress and the training regime forming robust memories.

Amounting research points to the utility of rat ultrasonic vocalizations to alter emotional states, evidenced by behavioral changes, in tested rats via playback of affectively valenced calls (*Bonauto et al., 2023*). We have exposed rats to 44-kHz playback along with 22-kHz and 50-kHz playback. The experimental design (see 'Materials and methods' for details) allowed us to compare rats' responses to 22-kHz vs. 44-kHz playbacks, especially with 50-kHz playback used as a form of control or baseline. In general, the rats responded similarly to hearing 44-kHz calls as they did to hearing aversive 22-kHz calls, especially regarding heart rate change, despite the 44-kHz calls occupying the frequency band of appetitive 50-kHz vocalizations. This is contrary to some observations (*Saito et al., 2019*), which suggested that frequency band plays the main role in rat ultrasound perception. Please factor in potential carry-over effects (resulting from hearing playbacks of the same valence in a row) in the differences between responses to 50-kHz vs. 22/44-kHz playbacks, especially those observed before the signal (*Figure 4A and B*). Other responses to 44-kHz calls were intermediate, they fell between response levels to appetitive vs. aversive playback, which might signify some behavioral specificity and importance (or possibly confusion). These latter effects were similar in both playback experiments despite an array of methodological differences between them. Overall, these initial results raise further questions about how, ethologically, animals may interpret the variation in hearing 22-kHz vs. 44-kHz calls and integrate this interpretation in their responses.

The question also is, why have the 44-kHz vocalizations been overlooked until now? On one hand, long (or not that long as in *Bialy et al., 2019*), frequency-stable high-pitch vocalizations have been reported before (e.g., *Sales, 1979*; *Shimoju et al., 2020*), notably as caused by intense cholinergic stimulation (*Brudzynski and Bihari, 1990*) or higher shock-dose fear conditioning (*Wöhr et al., 2005*). However, they have not been systematically defined, described, fully shown, or demonstrated to be a separate type of vocalization. On the other hand, 44-kHz calls were likely omitted as the analyses were restricted to canonical groups, that is, flat 22-kHz and short 50-kHz calls, with a sharp dividing frequency border between the two (e.g., *Kalamari et al., 2021*; *Potasiewicz et al., 2020*; *Turner et al., 2019*) or even a frequency 'safety gap' between 22-kHz and 50-kHz vocalizations (e.g., *Silkstone and Brudzynski,*

*2019*; *Garcia et al., 2015*). Moreover, many older bat detectors had limited frequency-range detection (e.g., up to 40-kHz in *Sales, 1991*) when stress-evoked types of ultrasonic calls were being established. Finally, 44-kHz vocalizations are emitted much fewer than 22-kHz calls (*Figure 1F and G*).

Here, we present introductory evidence that 44-kHz vocalizations are a separate and behaviorally relevant group of rat ultrasonic calls. These results require further confirmations and additional experiments, also in form of replication, including research on female rat subjects. However, our results bring to awareness that rats employ these previously unrecognized, long, high-pitched and flat aversive calls in their vocal repertoire. Researchers investigating rat ultrasonic vocalizations should be aware of their potential presence and to not rely fully on automated detection of high- vs. low-pitch calls.

## Materials and methods

**Key resources table**

| Reagent type (species) or resource | Designation | Source or reference | Identifiers | Additional information |
|---|---|---|---|---|
| Strain, strain background (Wistar rat, *Rattus norvegicus*) | Wistar rat | The Center for Experimental Medicine of the Medical University of Białystok | CMDB:WI | 7 weeks old upon arrival at our lab |
| Strain, strain background (SHR, *Rattus norvegicus*) | SHR | Mossakowski Medical Research Institute | SHR/NHsd/Cmd | 7 weeks old upon arrival at our lab |
| Strain, strain background (Sprague–Dawley Rat, *R. norvegicus*) | SD rat | Mossakowski Medical Research Institute | Tac:Cmd:SD | 7 weeks old upon arrival at our lab |
| Software, algorithm | SASLab Pro | Avisoft Bioacoustics | RRID:SCR_014438 | version 5.2.xx |
| Software, algorithm | Avisoft Recorder | Avisoft Bioacoustics | RRID:SCR_014436 | version 4.2.28 |
| Software, algorithm | Deep Squeak | https://github.com/DrCoffey/DeepSqueak; *Coffey and Marx, 2022* | RRID:SCR_021524 | version 3.0.4 |
| Software, algorithm | sklearn | PMID:24600388 | RRID:SCR_002577 | |
| Other, hardware | Ultrasonic Microphone | Avisoft Bioacoustics | CM16/CMPA | |
| Other, hardware | Radiotelemetric transmitters | Data Sciences International | HD-S10 | |

### Animals

Wistar rats (n = 167) were obtained from The Center for Experimental Medicine of the Medical University of Białystok, Poland; SHR (n = 80) and Sprague–Dawley rats (n = 16) were from Mossakowski Medical Research Institute, Polish Academy of Sciences, Poland. All rats were males, 7 weeks of age on arrival, randomly assigned into groups and cage pairs where appropriate; housed with a 12-h light–dark cycle, ambient temperature (22–25°C), with standard chow and water provided ad libitum. The animals were left undisturbed for at least 1 week before any procedures, then handled at least four times for 2 min by each experimenter directly involved for 1–2 weeks. All procedures were approved by the Local Ethical Committees for Animal Experimentation in Warsaw.

### Animal details: Groups of animals used

#### Trace fear conditioning experiment

Wistar rats, both single-housed (n = 14) and pair-housed (n = 20), were implanted with radiotelemetric transmitters for measuring heart rate in an ultrasonic vocalization playback experiment previously described by us (*Olszyński et al., 2020*) after which, at 13 weeks of age, half of them (n = 17) were fear-conditioned (10 shocks), while the other half (n = 17) served as controls (*Table 1*/Exp. 1/#1–4, n = 34).

#### Delay fear conditioning experiment, rats with transmitters

Wistar rats (n = 94) and SHR (n = 80) were implanted with radiotelemetric transmitters 1 week before fear conditioning during which they received 0, 1, 6, or 10 shocks at 12 weeks of age (*Table 1*/Exp. 2/#5–12, n = 174). All the details are described in *Olszyński et al., 2021* and *Olszyński et al., 2023*.

#### Delay fear conditioning experiment, rats without transmitters

Wistar rats were housed in pairs; were not implanted with radiotelemetric transmitters to eliminate the potential effect of surgical intervention on vocalization; they received 10 conditioning stimuli at

12 weeks of age (*Table 1*/Exp. 3/#13, n = 10) – the same as in *Olszyński et al., 2021* and *Olszyński et al., 2023*.

## Playback experiment, rats with transmitters

Wistar rats (n = 29) were housed in pairs; all were implanted with a radiotelemetric transmitter 1 week before the playback experiment. At 12 weeks of age, one group (n = 13) heard 50-kHz appetitive vocalization playback while the other (n = 16) 22-kHz and 44-kHz aversive calls (for details, see below).

## Playback experiment, rats without transmitters

Sprague–Dawley rats (n = 15) were housed in pairs, were not implanted with the transmitters, and received 22-kHz, 44-kHz, and 50-kHz ultrasonic vocalization playback at 8 weeks of age (see below).

## Surgery, transmitter implantation, and heart rate registration

Radiotelemetric transmitters (HD-S10, Data Sciences International, St. Paul, MN) were implanted into the abdominal aorta of rats in specified groups as previously described (*Olszyński et al., 2020*; *Olszyński et al., 2021*). An illustrative image with the surgery details can be found elsewhere (Figure 5 in *Pestana-Oliveira et al., 2020*; please note that tissue glue was used instead of cellulose patches and silk sutures). The signal was collected by receivers (RSC-1, Data Sciences International) as previously described (*Olszyński et al., 2020*; *Olszyński et al., 2021*; *Olszyński et al., 2023*). Readings were processed using Dataquest ART (version 4.36, Data Sciences International) for trace fear conditioning (*Table 1*/Exp. 1) and Ponemah (version 6.32, Data Sciences International) software for other experiments *Table 1*/Exp. 2–3 and playback experiments.

## Fear conditioning

All conditioning procedures were conducted in a chamber (VFC-008-LP, Med Associates, Fairfax, VT) located in an outer cubicle (MED-VFC2-USB-R, Med Associates) equipped with an ultrasound CM16/CMPA condenser microphone (Avisoft Bioacoustics, Berlin, Germany). Ultrasonic vocalizations were recorded via Avisoft USGH Recorder (Avisoft Bioacoustics), and rat behavior was recorded via NIR monochrome camera (VID-CAM-MONO-6, Med Associates). All procedures were described in detail before (*Olszyński et al., 2021*; *Olszyński et al., 2023*).

### Trace fear conditioning

Trace fear conditioning (*Table 1*/Exp. 1/#1–4, n = 34 rats) was performed similarly to some previous reports (e.g., *Jaholkowski et al., 2009*). Rats were individually placed in the fear conditioning apparatus in one of two different contexts: A (safe) or B (unsafe). Context A was in an illuminated room with the cage interior with white light, the cage floor was made of solid plastic, and the cage was scented with lemon odor, cleaned with a 10% ethanol solution; the experimenter was male wearing white gloves. Context B was a different, dark room, with the cage interior with green light, the floor was made of metal bars, and the cage was scented with mint odor, cleaned with 1% acetic acid; the experimenter was female with violet gloves. The procedure: on day –2, each rat was habituated to context A for 20 min; on day –1, habituated to context B for 20 min; on day 0, each rat was placed for 52 min in context A; on day 1, after 10 min in context B, the rat received 10 conditioning stimuli (15-s-long sine wave tone, 5-kHz, 85 dB) followed by a 30 s trace period and a foot-shock (1 s, 1 mA) and 210 s ITI; total session duration: 52 min. Control rats were subjected to the same procedures but did not receive the electric shock at the end of trace periods. The animals were tested with the same protocol without shocks in context A (day 2) and context B (day 3). During the test session, control animals showed a lower level of freezing than conditioned animals (1.3 ± 0.8% vs 19.7 ± 4.3% during the first 5 min in unsafe context B and 0.4 ± 0.3% vs 9.9 ± 1.9% during 10 s following the time of expected shock in context B, results averaged from the first 3 out of 10 trials; p=0.0003 and p=0.0001,

respectively, Mann–Whitney); none of the control animals emitted 44-kHz calls, neither the fear conditioning day nor the test days.

## Delay fear conditioning

(*Table 1*/Exp. 2–3/#5–13, n = 184 rats) The procedure and its results were described before (*Olszyński et al., 2021*; *Olszyński et al., 2023*); rats received 1, 6, or 10 conditioning stimuli (20-s-long white light co-terminating with an electric foot-shock, 1 s, 1 mA). For control rats, an equal time-length procedure was done for each conditioning protocol, that is, the same parameters as in 1, 6, or 10 stimuli groups, with no shock. Control animals showed a lower level of freezing than conditioned animals. There were only four ultrasonic calls we classified as 44-kHz vocalizations among 4126 vocalizations emitted by the control rats during training and testing. We did not observe any difference in the number of 44-kHz vocalizations between Wistar rats with transmitters vs. without transmitters during delay conditioning training (p=0.8642, Mann–Whitney). These two groups were therefore reported together.

## Measuring freezing

Freezing behavior was scored automatically using Video Freeze software (Med Associates) with a default motion index threshold of 18. To avoid including brief moments of the animal's stillness, freezing was measured only if the animal did not move for at least 1 s, that is, 30 video frames, with some exceptions, see next.

### Vocalization-nested freezing behavior

Freezing at the exact times of ultrasonic calling was measured in rats that had undergone 10 trials of fear conditioning training, which produced 44-kHz calls (n = 32, selected from *Table 1*/Exp. 1–3/#2,4,8,13). From each rat, one 44-kHz call was randomly selected along with the long 22-kHz call closest to it. Such pairs of vocalizations were selected with either ≥150 ms duration (n = 32) or ≥500 ms duration (n = 28). For each pair of vocalizations, the freezing behavior was calculated from the entire duration of the shorter call and for the equal-time-length period in the middle of the longer vocalization. Due to the shortened time scale, the minimal freezing detection window was reduced to 3 frames for ≥150-ms-long calls as well as 5, 10, and 15 frames – for ≥500 ms calls.

## Ultrasonic playback

It was performed as described previously (*Olszyński et al., 2020*; *Olszyński et al., 2021*; *Olszyński et al., 2023*) in individual experimental cages with acoustic stimuli presented through a Vifa ultrasonic speaker (Avisoft Bioacoustics) connected to an UltraSoundGate Player 116 (Avisoft Bioacoustics). Ultrasonic vocalizations emitted by the rat were recorded with a CM16/CMPA condenser microphone (Avisoft Bioacoustics). Both playback and recording of calls were performed using Avisoft Recorder USGH software (version 4.2.28, Avisoft Bioacoustics). The locomotor activity was recorded with an acA1300-60gc camera (Basler AG, Ahrensburg, Germany). Eight sets of ultrasonic vocalizations were presented:

1. **44-kHz long calls**, 8 calls in one repeat, constant frequency (2.7 ± 0.1 kHz max–min frequency difference), 42.1 ± 0.2 kHz peak frequency, 1064.3 ± 89.6 ms duration with 199.0 ± 14.7 ms sound intervals.
2. **22-kHz long calls,** 8 calls in one repeat, typical long 22-kHz vocalizations, constant frequency (1.9 ± 0.9 kHz max–min frequency difference), 24.5 ± 0.2 kHz peak frequency, 1066.4 ± 90.2 ms duration with 195.6 ± 15.5 ms sound intervals.
3. **22-kHz short modulated calls**, 26 calls in two repeats, short (<300 ms), not resembling typical 22-kHz long calls (5.3 ± 0.4 kHz max–min frequency difference), 22.7 ± 0.6 kHz peak frequency, 24.7 ± 1.6 ms duration with 172.8 ± 5.6 ms sound intervals.
4. **22-kHz short flat calls**, 43 calls in one repeat, short (<300 ms), resembling typical 22-kHz long calls, constant frequency (2.3 ± 0.1 kHz max–min frequency difference), 25.1 ± 0.3 kHz peak frequency, 102.4 ± 10.9 ms duration with 132.1 ± 6.2 ms sound intervals.
5. **50-kHz modulated calls**, 23 calls in two repeats, moderately modulated (8.6 ± 0.3 kHz max–min frequency difference), 61.0 ± 0.8 kHz peak frequency, 37.6 ± 1.5 ms duration with 183.7 ± 4.5 ms sound intervals.

6. **50-kHz flat calls**, 29 calls in two repeats, constant frequency (4.2 ± 0.2 kHz max–min frequency difference), 53.5 ± 0.5 kHz peak frequency, 66.2 ± 3.8 ms duration with 144.1 ± 4.4 ms sound intervals.
7. **50-kHz trill calls**, 29 calls in two repeats, highly modulated (37.4 ± 1.7 kHz max–min frequency difference), 68.0 ± 0.9 kHz peak frequency, 53.7 ± 1.4 ms duration with 158.5 ± 4.9 ms sound intervals.
8. **50-kHz mixed calls**, used previously in *Olszyński et al., 2020*, *Olszyński et al., 2021*, and *Olszyński et al., 2023*, 28 calls, in three repeats, frequency modulated and trill subtypes, 9.8 ± 1.9 kHz max–min frequency difference, 58.6 ± 0.7 kHz peak frequency, 28.4 ± 1.6 ms duration with 91.4 ± 1.4 ms sound intervals.

Calls were presented with a sampling rate of 250 kHz in 16-bit format. All calls except for 50-kHz mixed calls were collected in our laboratory from fear conditioning or playback experiments. Calls in the same set were taken from one animal wherever possible. The sound interval was adjusted if it was peculiarly long or the sequence was interrupted by other types of calls in the original recordings.

## Playback procedure, rats with transmitters
As previously described (*Olszyński et al., 2020*; *Olszyński et al., 2021*; *Olszyński et al., 2023*). Before playback presentation, animals were habituated for 3 min to the experimental conditions, that is, recording cage, presence of the speaker and microphone, over 4 days. Habituated rats then underwent a playback procedure; in short, after 10 min of silence, the rats were exposed to four 10-s-long call sets (either aversive or appetitive) with 5-min-long ITI in-between; a rat that received appetitive playback was followed by a rat receiving aversive playbacks, etc. Also, the order of the presented sets was randomized between animals. The aversive-calls playback contained sets nos. 1–4. The appetitive-calls playback contained sets nos. 5–8. Since initial analysis showed no differences within responses to 22-kHz aversive sets and within responses to 50-kHz appetitive sets, we decided to show the results following playback of 44-kHz long calls (set no. 1), 22-kHz long calls (set no. 2), and 50-kHz modulated calls (set no. 5) only.

## Playback procedure, rats without transmitters
Before playback presentation, animals were habituated for 3 min to the experimental conditions, that is, recording cage, presence of the speaker and microphone, over 4 days. After 5 min of initial silence, the rats were presented with two 10-s-long playback sets of either 22-kHz (set no. 2; n = 7) or 44-kHz calls (set no. 1; n = 8), followed by one 50-kHz modulated call 10-s set (no. 5) and another two playback sets of either 44-kHz or 22-kHz calls not previously heard. The playback presentations were separated by 3 min ITI. Responses to the pairs of playback sets were averaged.

## Locomotor activity in playback
An automated video tracking system (Ethovision XT 10, Noldus, Wageningen, The Netherlands) was used to measure the total distance traveled (cm). Proximity to the speaker was expressed as the percentage of time spent in the half of the cage closer to the ultrasonic speaker. The center point of each animal's shape was used as a reference point for measurements of locomotor activity, thus registering only full-body movements.

## Analysis of ultrasonic vocalizations
Audio recordings were analyzed manually using SASLab Pro (version 5.2.xx, Avisoft Bioacoustics) as described (*Dupin et al., 2019*; *Olszyński et al., 2020*; *Olszyński et al., 2021*; *Olszyński et al., 2023*) to measure key features of calls and categorize them into subtypes.

## Sound mean power
This was measured as the average spectra power density of the vocalization contour using DeepSqueak software. Initially, calls were detected using the default rat long-vocalization neural network (Long Rat Detector YOLO R1) and subsequently manually reviewed and corrected where necessary. We analyzed a subset of Wistar rats subjected to 10-trial fear conditioning training that emitted more than 20 instances of 44-kHz calls during the fear conditioning training session (n = 17, selected from *Table 1*/Exp. 1–3/#2,4,8,13). It is important to note that due to the directional characteristics of the

microphones used, angular attenuation occurred during audio recording. This phenomenon results in a selective reduction in the intensity of higher frequency sounds, dependent on the angle between the sound emitter and the microphone (as specified in the CM16/CMPA microphone hardware specification page, *Avisoft Bioacoustics, 2023*). In our experimental setup, we approximated a 45° angle between the plane of the rat's head and the plane of the microphone's membrane. This angle corresponds to an estimated 10 dB attenuation (adopting a conservative estimate) of 40-kHz frequencies compared to 20-kHz frequencies for which there is even a small dB gain due to these hardware properties, 44-kHz calls are predicted to be approximately at least 10 dB louder in reality than what was recorded.

## 22-kHz vs. 44-kHz frequency ratio

A clear transition between 22-kHz and 44-kHz long calls was observed in n = 13 Wistar rats and n = 1 SHR. In each case, ten 22-kHz calls followed by ten 44-kHz calls were analyzed (n = 14, selected from *Table 1*/Exp. 1–3/#2,4,6–8,10–13).

## Step up and step down frequency ratio

Rats that emitted at least five vocalizations of the specific subtype were analyzed (step up, n = 14; step down, n = 13; selected from *Table 1*/Exp. 1–3/#2,4,7,8,13); five calls of the two subtypes from each rat were chosen randomly and the frequencies of their elements were measured.

## Ultrasonic vocalizations clustering (two independent methods)

Calls of conditioned and control animals were taken from all fear conditioning training sessions (*Table 1*/Exp. 1–3, n = 218). We used DBSCAN algorithm (*Ester et al., 1996*), a density based method, from the scikit-learn (sklearn) Python package, because of its ability to detect a desired number of clusters of arbitrary shape; with two main input parameters: MinPts (minimal number of points forming the core of the cluster) and ε (the maximum distance two points can be from one another while still belonging to the same cluster). To avoid detecting small clusters, we limited MinPts to 150 samples. The heuristic method described by *Ester et al., 1996* was implemented to find the initial range of $\varepsilon$. All the input data were standardized. The silhouette coefficient (*Rousseeuw, 1987*) was used to control the quality of the clustering. Maximizing $\varepsilon$ among different ranges helped to select the most relevant number of identified clusters. Clustering with $\varepsilon$ in the range of 0.14–0.2 resulted in a silhouette coefficient around 0.2–0.5.

## k-means algorithm

Vocalizations of selected fear-conditioned rats with 6–10 shocks and >30 of 44-kHz calls (n = 26, selected from *Table 1*/Exp. 1–3/#2,4,7,8,11–13) were detected using a built-in neural network for long rat calls (Long Rat Detector YOLO R1) on DeepSqueak (*Coffey et al., 2019*) software (version 3.0.4) running under MATLAB (version 2021b, MathWorks, Natick, MA) and manually revised for missed and mismatched calls. Unsupervised k-means clustering was based on call contour, frequency and duration variables, with equal weights assigned, and several descending elbow optimization parameters were used to obtain different maximum numbers of clusters together with Uniform Manifold Approximation and Projection for Dimension Reduction (UMAP) (*McInnes et al., 2018*) for superimposing and visualization of clusters.

## Quantification and statistical analysis

Data were analyzed using non-parametric Friedman, Wilcoxon, Mann–Whitney tests with GraphPad Prism 8.4.3 (GraphPad Software, San Diego, CA); the p values are given, p<0.05 as the minimal level of significance. In particular, the Friedman test was used to assess the presence of change within the sequence of several ITI, while the Wilcoxon test was used for the difference between the first and the last ITI analyzed. Figures were prepared using the same software and depict average values with a standard error of the mean (SEM).

## Acknowledgements

We thank Iryna Artemieva for her help with DeepSqueak analysis. We would also like to thank Patrick Reilly and Adelaide Yiu for their advice and assistance. This research was funded by the National Science Centre, Poland, grant OPUS no. 2015/19/B/NZ4/03393 (RKF) and by Mossakowski Medical Research Institute, PAS, Poland, Internal Research Fund no. FBW-17 (RKF). RP and ADW were supported by European Social Fund, POWR.03.02.00-00-I028/17-00.

## Additional information

### Funding

| Funder | Grant reference number | Author |
| --- | --- | --- |
| National Science Centre, Poland | 2015/19/B/NZ4/03393 | Robert Kuba Filipkowski |
| Mossakowski Medical Research Institute, Polish Academy of Sciences | FBW-17 | Robert Kuba Filipkowski |
| European Social Fund | POWR.03.02.00-00-I028/17-00 | Rafał Polowy Agnieszka Diana Wardak |

The funders had no role in study design, data collection and interpretation, or the decision to submit the work for publication.

### Author contributions

Krzysztof Hubert Olszyński, Conceptualization, Resources, Data curation, Formal analysis, Supervision, Validation, Investigation, Visualization, Writing – original draft, Project administration, Writing – review and editing; Rafał Polowy, Conceptualization, Data curation, Software, Formal analysis, Validation, Investigation, Visualization, Writing – original draft, Writing – review and editing; Agnieszka Diana Wardak, Formal analysis, Investigation, Writing – review and editing; Izabela Anna Łaska, Formal analysis, Writing – review and editing; Aneta Wiktoria Grymanowska, Olga Gawryś, Investigation, Writing – review and editing; Wojciech Puławski, Formal analysis, Writing – review and editing, Investigation; Michał Koliński, Formal analysis, Visualization, Writing – review and editing; Robert Kuba Filipkowski, Conceptualization, Resources, Supervision, Funding acquisition, Validation, Visualization, Methodology, Writing – original draft, Project administration, Writing – review and editing

### Author ORCIDs

Krzysztof Hubert Olszyński ⓘ https://orcid.org/0000-0003-2114-4487
Rafał Polowy ⓘ https://orcid.org/0000-0001-6472-3695
Agnieszka Diana Wardak ⓘ https://orcid.org/0000-0003-1799-8949
Izabela Anna Łaska ⓘ https://orcid.org/0009-0002-2861-6125
Aneta Wiktoria Grymanowska ⓘ https://orcid.org/0009-0005-9240-4532
Wojciech Puławski ⓘ https://orcid.org/0000-0002-4837-1148
Olga Gawryś ⓘ https://orcid.org/0000-0002-4397-3991
Michał Koliński ⓘ https://orcid.org/0000-0003-1047-2186
Robert Kuba Filipkowski ⓘ https://orcid.org/0000-0002-9911-9751

### Ethics

Procedures involving rats were conducted in accordance with the Declaration of Helsinki, and approved by the First Warsaw Local Ethics Committee for Animal Experimentation (646/2018) as well as the Second Warsaw Local Ethics Committee for Animal Experimentation (WAW2/093/2019).

Reviewer #2 (Public Review): https://doi.org/10.7554/eLife.88810.4.sa1
Author response https://doi.org/10.7554/eLife.88810.4.sa2

# Additional files

## Supplementary files
• MDAR checklist

## Data availability

Raw data (e.g., all calls' peak frequency and duration) analyzed, data supporting clustering files for DBSCAN (.csv), extracted call contours for k-means (.mat), ultrasonic playback files used (.wav), and two selected sample recordings of ultrasonic vocalizations, registered during fear conditioning session have been deposited to Mendeley Data. Due to the large size of the .wav files, the remaining recordings will be available upon request, provided that file hosting is arranged by the requesting party.

The following dataset was generated:

| Author(s) | Year | Dataset title | Dataset URL | Database and Identifier |
|---|---|---|---|---|
| Filipkowski RK | 2024 | Male rats emit aversive 44-kHz ultrasonic vocalizations during prolonged Pavlovian fear conditioning. Olszyński, Polowy et al. | https://data.mendeley.com/datasets/ypxwmsh8hr/1 | Mendeley Data, 10.17632/ypxwmsh8hr.1 |

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
