## [Editor Report · eLife Assessment]

This **valuable** study investigated the appearance of ultrasonic vocalizations around 44 kHz that occurs in response to prolonged fear conditioning in male rats. Evidence in support of the conclusions is **solid** and may be of interest to some researchers also investigating distress-related ultrasonic vocalizations.

---

## [Referee Report · Reviewer #2 (Public Review)]

Olszyński et al. claim that they identified a "new-type" ultrasonic vocalization around 44 kHz that occurs in response to prolonged fear conditioning (using foot-shocks of relatively high intensity, i.e. 1 mA) in rats. Typically, negative 22-kHz calls and positive 50-kHz calls are distinguished in rats, commonly by using a frequency threshold of 30 or 32 kHz. Olszyński et al. now observed so-called "44-kHz" calls in a substantial number of subjects exposed to 10 tone-shock pairings, yet call emission rate was low (according to Fig. 1G around 15%, according to the result text around 7.5%). They also performed playback experiments and concluded that "the responses to 44-kHz aversive calls presented from the speaker were either similar to 22-kHz vocalizations or in-between responses to 22-kHz and 50-kHz playbacks".

Strengths: Detailed spectrographic analysis of a substantial data set of ultrasonic vocalizations recorded during prolonged fear conditioning, combined with playback experiments.

---

## [Author Response]

The following is the authors’ response to the previous reviews.

**Reviewer #1 (Public Review):**
The exclusive use of males is a major concern lacking adequate justification and should be disclosed in the title and abstract to ensure readers are aware of this limitation. With several reported sex differences in rat vocal behaviors this means caution should be exercised when generalizing from these findings. The occurrence of an estrus cycle in typical female rats is not justification for their exclusion. Note also that male rodents experience great variability in hormonal states as well, distinguishing between individuals and within individuals across time. The study of endocrinological influences on behavior can be separated from the study of said behavior itself, across all sexes. Similarly, concerns about needing to increase the number of animals when including all sexes are usually unwarranted (see Shansky [2019] and Phillips et al. [2023]).

As suggested by the Reviewer, we have disclosed the use of males in the title and the abstract. Also, we have added the statement that research on female rat subjects is required: “Here we are showing introductory evidence that 44-kHz vocalizations are a separate and behaviorally-relevant group of rat ultrasonic calls. These results require further confirmations and additional experiments, also in form of repetition, including research on female rat subjects.”

Regarding the analysis where calls were sorted using DBSCAN based on peak frequency and duration, my comment on the originally reviewed version stands. It seems that the calls are sorted by an (unbiased) algorithm into categories based on their frequency and duration, and because 44kHz calls differ by definition on frequency and duration the fact that the algorithm sorts them as a distinct category is not evidence that they are "new calls [that] form a separate, distinct group". I appreciate that the authors have softened their language regarding the novelty and distinctness of these calls, but the manuscript contains several instances where claims of novelty and specificity (e.g. the subtitle on line 193) is emphasized beyond what the data justifies.

We further softened our language regarding novelty and distinctness of 44-kHz vocalizations – including the aforementioned subtitle. However, in response, we would like to bring to the readers’ attention that all major groups of calls, i.e., long 22-kHz calls, short 22-kHz calls, and 50-kHz vocalization, are also defined in our manuscript and in the literature by their frequency and duration. However not one of these groups was identified separately by DBSCAN clustering excepting the 44kHz vocalizations. If they were not a distinct group, we would expect the 44-kHz and 50-kHz vocalizations to blend first (because of the similar frequencies) or 44-kHz and 22-kHz calls to merge first (because of the similar durations), but they do not in this unbiased examination.

The behavioral response to call playback is intriguing, although again more in line with the hypothesis that these are not a distinct type of call but merely represent expected variation in vocalization parameters. Across the board animals respond rather similarly to hearing 22 kHz calls as they do to hearing 44 kHz calls, with occasional shifts of 44 kHz call responses to an intermediate between appetitive and aversive calls. This does raise interesting questions about how, ethologically, animals may interpret such variation and integrate this interpretation in their responses. However, the categorical approach employed here does not address these questions fully.

This paragraph is exactly the same as in the previous review. There was no comment regarding our previous answer. Here is the previous answer:

“We are unsure of the Reviewer’s critique in this paragraph and will attempt to address it to the best of our understanding. Our finding of up to >19% of long seemingly aversive, 44-kHz calls, at a frequency in the define appetitive ultrasonic range (usually >32 kHz) is unexpected rather than “expected”. We would agree that aversive call variation is expected, but not in the appetitive frequency range.

Kindly note the findings by Saito et al. (2019), which claim that frequency band plays the main role in rat ultrasonic perception. It is possible that the higher peak frequency of 44-kHz calls may be a strong factor in their perception by rats, which is, however, modified by the longer duration and the lack of modulation.

Also, from our experience, it is quite challenging to demonstrate different behavioral responses of naïve rats to pre-recorded 22-kHz (aversive) vs. 50-kHz (appetitive) vocalizations. Therefore, to demonstrate a difference in response to two distinct, potentially aversive, calls, i.e., 22kHz vs. 44-kHz calls, to be even more difficult (as to our knowledge, a comparable experiment between short vs. long 22-kHz ultrasonic vocalizations, has not been done before).

Therefore, we do not take lightly the surprising and interesting finding that “animals respond rather similarly to hearing 22 kHz calls as they do to hearing 44 kHz calls, with occasional shifts of 44 kHz call responses to an intermediate between appetitive and aversive calls”. We would rather put this description in analogous words: “the rats responded similarly to hearing 44-kHz calls as they did to hearing aversive 22-kHz calls, especially regarding heart-rate change, despite the 44-kHz calls occupying the frequency band of appetitive 50-kHz vocalizations” and “other responses to 44-kHz calls were intermediate, they fell between response levels to appetitive vs. aversive playback” – which we added to the Discussion.

Finally, we acknowledge that our findings do not present a finite and complete picture of the discussed aspects of behavioral responses to the presented ultrasonic stimuli (44-kHz vocalizations). Therefore, we have incorporated the Reviewer’s suggestion in the discussion. The added sentence reads: “Overall, these initial results raise further questions about how, ethologically, animals may interpret the variation in hearing 22-kHz vs. 44-kHz calls and integrate this interpretation in their responses.”

I appreciate the amendment in discussing the idea of arousal being the key determinant for the increased emission of 44kHz, and the addition of other factors. Some of the items in this list, such as annoyance/anger and disgust/boredom, don't really seem to fit the data. I'm not sure I find the idea that rats become annoyed or disgusted during fear conditioning to be a particularly compelling argument. As such the list appears to be a collection of emotion-related words, with unclear potential associations with the 44kHz calls.

We agree that most of the factors listed are not supported by the data. These are hypotheses and speculations only – hence, an assumption / tentative statement, i.e., “It could also be argued that…”. We have changed it into “It could also be speculated that…”.

Later in the Discussion the authors argue that the 44kHz aversive calls signal an increased intensity of a negative valence emotional state. It is not clear how the presented arguments actually support this. For example, what does the elongation of fear conditioning to 10 trials have to do with increased negative emotionality? Is there data supporting this relationship between duration and emotion, outside anthropomorphism? Each of the 6 arguments presented seems quite distant from being able to support this conclusion.

We have added a description summarizing the literature that expounds the differences in employing one-two vs. five-ten foot-shocks during fear-conditioning training. It says:

“Importantly, it has been demonstrated multiple times that training rats with several electric foot-shocks (i.e., 5-10 shocks) produces a qualitatively different kind of fear-memory compared to training with only 1-2 shocks. Training with more numerous shocks has been shown to result in augmented freezing (e.g., Fanselow and Bolles, 1979, Haubrich et al., 2020, Haubrich and Nader, 2023, Poulos et al., 2016, Wang et al., 2009) which reflects a more intense fear-memory that is resistant to extinction (Haubrich et al., 2020, Haubrich and Nader, 2023), resistant to reconsolidation blockade (Haubrich et al., 2020, Wang et al., 2009, Finnie and Nader, 2020), associated with downregulation of NR2B NMDA-receptor subunits as well as elevated amyloid-beta concentrations in the lateral and basal amygdala (Finnie and Nader, 2020, Wang et al., 2009). Additionally, it involves activation of the noradrenaline-locus coeruleus system (Haubrich et al., 2020) and collective changes in connectivity across multiple brain regions within the neural network (Haubrich and Nader, 2023).

Notably, it has also been shown that higher freezing as a result of fear-conditioning training correlates with increased concentrations of stress hormone, corticosterone, in the blood (Dos Santos Correa et al., 2019). The rats subjected to 6- and 10-trial fear conditioning, whose results are reported herein (Tab. 1/Exp. 2/#7,8,11,12; n = 73), also demonstrated higher freezing than rats subjected to 1trial conditioning (Tab. 1/Exp. 2/#6,10; n = 33), which is reported elsewhere (Olszynski et al., 2021, Fig. S1C-E; Olszynski et al., 2022, Fig. S1D-G). Therefore, we postulate that emission of 44-kHz calls is associated with increased stress and the training regime forming robust memories.”

In sum, rather than describing the 44kHz long calls as a new call type, it may be more accurate to say that sometimes aversive calls can occur at frequencies above 22 kHz. Individual and situational variability in vocalization parameters seems to be expected, much more so than all members of a species strictly adhering to extremely non-variable behavioral outputs.

This paragraph is exactly the same as in the previous review. There was no comment regarding our previous answer. Here is the previous answer:

“The surprising fact that there are presumably aversive calls that are beyond the commonly applied thresholds, i.e., >32 kHz, while sharing some characteristics with 22-kHz calls, is the main finding of the current publication. Whether they be finally assigned as a new type, subtype, i.e. a separate category or become a supergroup of aversive calls with 22-kHz vocalizations is of secondary importance to be discussed with other researchers of the field of study.

However, we would argue – by showing a comparison – that 22-kHz calls occur at durations of <300 ms and also >300 ms, and are, usually, referred to in literature as short and long 22-kHz vocalizations, respectively (not introduced with a description that “sometimes 22-kHz calls can occur at durations below 300 ms”). These are then regarded and investigated as separate groups or classes usually referred to as two different “types” (e.g., Barker et al., 2010) or “subtypes” (e.g., Brudzynski, 2015). Analogously, 44-kHz vocalizations can also be regarded as a separate type or a subtype of 22kHz calls. The problem with the latter is that 22-kHz vocalizations are traditionally and predominantly defined by 18–32 kHz frequency bandwidth (Araya et al., 2020; Barroso et al., 2019; Browning et al., 2011; Brudzynski et al., 1993; Hinchcliffe et al., 2022; Willey & Spear, 2013).”

**Reviewer #1 (Recommendations For The Authors):**
Additional considerations:Abstract: The 19.4% seems to be the percentage of 44 kHz calls observed during the 9th trial of the 10trial experiment, not the percentage of calls that were 44kHz during bouts of freezing.

We clarified the sentence. It now says:

“We observed 44-kHz calls to be associated with freezing behavior during fear conditioning training, during which they constituted up to 19.4% of all calls and most of them appeared next to each other forming groups of vocalizations (bouts).”

Abstract: "We hope that future investigations of 44-kHz calls in rat models of human diseases will contribute to expanding our understanding and therapeutic strategies related to human psychiatric conditions." This sounds like a far too strong of an implication provided the link between these calls and models of human psychiatric conditions is not clear.

We agree, the link is not clear. Therefore we only express our hope. We hope “the link” is there. While other ultrasonic calls are already being investigated in such animal models, training regimes employing numerous electric shocks are used as models of PTSD, helplessness etc.

Line 101: Seems a strong assumption to state the authors of the other publication were inspired by this paper, unless there is personal communication corroborating this.

The wording of the sentence has been changed.

It is still not clear why both Friedman and Wilcoxon tests were used, especially in situations where only one result seems to be referenced (for example on line 108-109).

We added the explanation within Methods: “In particular, the Friedman test was used to assess the presence of change within the sequence of several ITI, while the Wilcoxon test was used for the difference between the first and the last ITI analyzed.”